# Developing New Tools to Fight Human Pathogens: A Journey through the Advances in RNA Technologies

**DOI:** 10.3390/microorganisms10112303

**Published:** 2022-11-21

**Authors:** Vanessa G. Costa, Susana M. Costa, Margarida Saramago, Marta V. Cunha, Cecília M. Arraiano, Sandra C. Viegas, Rute G. Matos

**Affiliations:** Instituto de Tecnologia Química e Biológica António Xavier, Universidade Nova de Lisboa, Avenida da República, 2780-157 Oeiras, Portugal; vanessa.costa@itqb.unl.pt (V.G.C.); smma.costa@itqb.unl.pt (S.M.C.); margaridasaramago@itqb.unl.pt (M.S.); mvcunha@itqb.unl.pt (M.V.C.); cecilia@itqb.unl.pt (C.M.A.)

**Keywords:** ribonucleases, small non-coding RNAs (ncRNAs), virulence, RNA metabolism, RNA chaperones, CRISPR, RNA regulators, synthetic biology, RNA tool

## Abstract

A long scientific journey has led to prominent technological advances in the RNA field, and several new types of molecules have been discovered, from non-coding RNAs (ncRNAs) to riboswitches, small interfering RNAs (siRNAs) and CRISPR systems. Such findings, together with the recognition of the advantages of RNA in terms of its functional performance, have attracted the attention of synthetic biologists to create potent RNA-based tools for biotechnological and medical applications. In this review, we have gathered the knowledge on the connection between RNA metabolism and pathogenesis in Gram-positive and Gram-negative bacteria. We further discuss how RNA techniques have contributed to the building of this knowledge and the development of new tools in synthetic biology for the diagnosis and treatment of diseases caused by pathogenic microorganisms. Infectious diseases are still a world-leading cause of death and morbidity, and RNA-based therapeutics have arisen as an alternative way to achieve success. There are still obstacles to overcome in its application, but much progress has been made in a fast and effective manner, paving the way for the solid establishment of RNA-based therapies in the future.

## 1. Introduction

A crucial characteristic of the prokaryotic world is its rapid ability to adjust to a changing environment. In the case of pathogenic organisms, it is also essential that they overcome the host immune system. This implies an extensive and prompt re-adjustment of the gene expression by complex regulatory networks, in which RNA metabolism has a crucial role. In fact, RNA is much more than a messenger as it is able to dynamically coordinate and instruct cellular functions, and it has also emerged as an important feature to be considered for the pathogenesis of microorganisms.

Bacterial infections are associated with a high rate of human morbidity and mortality, and bacterial resistance to antibiotics is an escalating problem worldwide. The widespread use of conventional antibiotics has favored the apperance of drug-resistant pathogens, and there is a growing need for the development of novel antibacterial strategies. The idea of directly targeting RNA is emerging as a new frontier in drug discovery studies, with the ultimate goal of expanding the antibiotic arsenal. The differences between the molecular machinery that governs bacterial and eukaryotic RNA metabolism are fundamental to identify in order to take advantage of this attractive drug target.

The stability of messenger RNA relies on several features and involves numerous players, with ribonucleases (RNases) being among the most important ones. These enzymes are ubiquitous and can perform the RNA degradation alone or in multiprotein complexes. The diversity of RNA molecules with regulatory roles is better understood now, including a wide range of small non-coding RNAs (ncRNAs) and the natural RNA interference (RNAi), CRISPR/Cas (Clustered Regularly Interspaced Short Palindromic Repeats/CRISPR-associated protein) and ERASE (Endogenous Reverse Transcriptase/RNase H-mediated Antiviral System) systems. The ERASE system [1] is a DNA-mediated RNA cleavage mechanism that is parallel to the RNA-guided DNA cleavage of the CRISPR/Cas system and the RNA-guided RNA cleavage of the RNAi pathway. The prospect of using this system to fight pathogenic infections has not yet been explored. All of the other RNA molecules and their therapeutics applications are further explained in the scope of this review.

Altogether, recent advances in RNA studies at a global scale have given us a vast amount of information about the role of the RNA regulation of pathogens. In this context, synthetic biology (SynBio) is emerging as a field that is focused on engineering biomolecular systems for a variety of applications. SynBio devices contribute not only to improve our understanding of the disease mechanisms, but also provide novel diagnostic tools. The developments in this field have created strategies for pathogen characterization, cancer treatment, vaccine development, microbiome engineering, cell therapy, regenerative medicine and the production of new and more affordable drugs. Additionally, the targeting of regulatory RNA-based interactions has broadened the SynBio applications in antimicrobial therapeutics. The inherent modularity and compatibility of RNA-based control components enables them to be independently optimized or exchanged, thus expanding their applications.

In this review, we discuss the connections between RNA metabolism and pathogenesis, uncovering how several techniques have helped to increase the knowledge in the field. Moreover, we cover some of the innovative SynBio systems in the area of the diagnosis and treatment of infectious diseases and the latest research on the usage of antisense antimicrobial therapeutics and CRISPR–Cas as a tool. Lastly, the present applications and the future prospects of mRNA vaccines are also examined. Overall, our main aim is to explore the emerging technologies in the RNA field and their application to current health problems.

## 2. Ribonucleases (RNases)

Ribonucleases (RNases) are the enzymes that determine the levels of functional RNAs in the cell, validate the quality control of all of the transcripts and allow the recycling of cellular ribonucleotides, which makes them key members of the RNA metabolism machinery [2]. Their diversity, structures, targets, and modes of action can vary significantly, providing multiple solutions for a similar outcome. These enzymes can be divided into endoribonucleases, which cleave the RNA molecules internally, and exoribonucleases, which degrade the RNA from one of its extremities [2,3]. Some of the existing RNases in the cell are essential enzymes, while others have overlapping functions. However, all of them operate according to the requirements of growth in adaptation to a specific environment, and they carry out surveillance. With this involvement in post-transcriptional mechanisms, ribonucleases have been associated with essential bacterial and viral processes (reviewed in [4,5,6]).

Tobe and colleagues showed that the product that was encoded by the gene *vacB* was required for the expression of several invasion factors in *Shigella flexneri,* and its deletion affected the bacterial capacity to adhere and spread inside host cells [7] (Figure 1). Later, this *vacB* gene was demonstrated to code for the exoribonuclease RNase R, and it was renamed *rnr* [8]. Other works have been published throughout the years, unravelling the impact of RNase R [9,10] and other ribonucleases in the different steps of the bacterial infection process (e.g., the exoribonucleases PNPase [11,12], and RNase AS [13] and the endoribonucleases RNase E [14], RNase III [14], YbeY [15], RNase J [16] and RNase Y [17]). These studies were performed in a plethora of bacterial species, including 10 of the 12 bacterial families that are considered to be ‘priority pathogens’ which pose the greatest threat to human health, according to the World Health Organization (WHO) [18].

RNases are found in all domains of life, and also in viruses. Viruses do not have their own metabolism, so many of them use host proteins during their life cycle, and they rarely code for RNases. When they are present, viral RNases are usually involved in specific steps of viral gene expression, genome replication, shutoff of host cell protein synthesis and host immune evasion, among others [19,20,21,22,23]. An example of viruses that code for RNases is coronaviruses. With the appearance of the coronavirus disease 2019 (COVID-19) pandemic, caused by SARS-CoV-2, viral ribonucleases have gained interest in the scientific community. Coronaviruses are RNA viruses that have in their replication machinery two crucial ribonucleases, nsp14 and nsp15, which are among the strongest interferon antagonists of SARS-CoV-2 [24,25,26] (Figure 2A). Nsp14 is a peculiar enzyme that harbors two distinct enzymatic activities, acting both as an exoribonuclease (ExoN) and as an N7-methyltransferase [27]. SARS-CoV-2 nsp14 ExoN knockout mutants are not viable [28,29]. The ExoN activity is responsible for the proofreading capacity of the viral genome during replication, a feature that has not been previously reported in any other RNA virus [28,30], and this activity is stimulated through the interaction with nsp10 [27]. Nsp15 is a conserved endoribonuclease specific of *Nidovirales* viruses, which plays fundamental roles in coronavirus pathogenesis and in the evasion of the host’s innate immune system [24,31,32].

**Figure 1 microorganisms-10-02303-f001:**
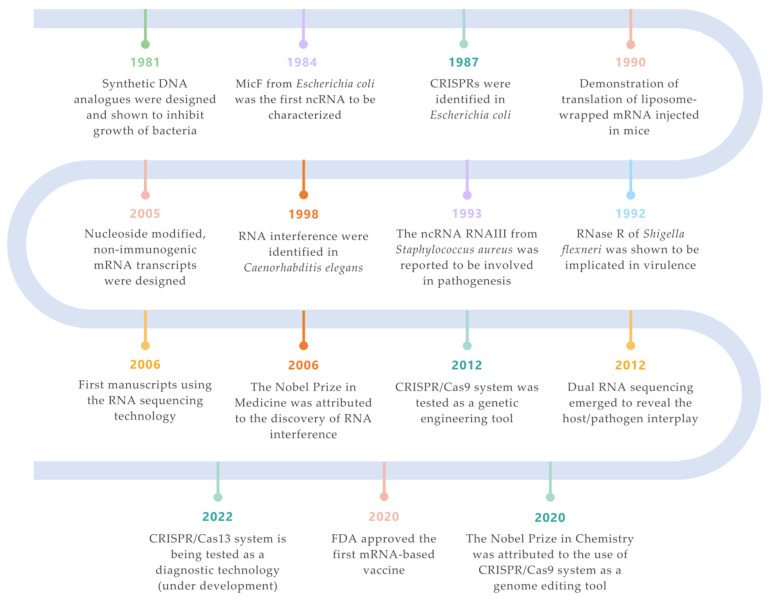
Landmarks on RNA technologies. An overview of the most relevant achievements and pioneer experiments around the RNA molecule. The timeline is color-coded for each field (green for synthetic biology; orange for siRNAs; purple for ncRNAs; turquoise for CRISPR/Cas systems; salmon for mRNA vaccines; yellow for RNA-seq technologies; blue for ribonucleases) [7,33,34,35,36,37,38,39,40,41,42,43].

Taken together, these ribonucleases, which play so many critical roles in both bacterial and viral processes, are very attractive targets for drug designs. The use of small molecules to inhibit the enzymatic activity of these proteins was already reported in *Staphylococcus aureus*, *Escherichia coli*, *Mycobacterium tuberculosis*, and SARS-CoV-2. In 2011, Olson et al. discovered a small molecule inhibitor of the protein component of the *S. aureus* ribonuclease P, and this inhibitor exhibited antimicrobial activity even against predominant antibiotic-resistant lineages [44]. A few years later, Kenneth McDowall and his team described a method of selecting small molecule inhibitors against RNase E, an essential *E. coli* endoribonuclease. These inhibitors were also demonstrated to be effective against the endoribonuclease RNase G in vitro (a protein that is homologous to the catalytic domain of RNase E). Additionally, they were shown to bind and inhibit the catalysis of an *M. tuberculosis* homologue, thus demonstrating a wider application of these inhibitors [45].

Recently, several in silico studies have proposed drug candidates that could inhibit SARS-CoV-2 ribonucleases (Figure 2A). From these, only a minor fraction was tested in vitro regarding their ability to affect the ribonucleolytic activity, and as a consequence, viral replication. For instance, the mycotoxin patulin was described as a specific inhibitor of the SARS-CoV-2 nsp14 ExoN activity, but it only decreased cell viability in vitro when it was used in high concentrations [46]. Disulfiram/Ebselen leads to the inhibition of the ExoN activity of nsp14, and in combination with Remdesivir, it can synergistically inhibit SARS-CoV-2 replication in Vero E6 cells [47]. Tipiracil, which is currently being used in cancer treatment, was shown to inhibit the endonuclease activity of nsp15 in vitro, but an improvement of the compound affinity is needed for it to serve as an antiviral drug in vivo [48]. Dutasteride, Meprednisone and Tasosartan were also able to inhibit nsp15 activity in vitro [49]. A clinical trial using Dutasteride has already shown its beneficial effect in the treatment of COVID-19 [50].

## 3. Small Non-Coding RNAs (ncRNAs or sRNAs)

Small non-coding RNAs are ubiquitous in bacterial species, and they are essential for their adaptation and survival under stress [51]. These transcripts are generally short, being about 50 to 500 nucleotides in length, usually highly structured, and they have the capacity to alter gene expression affecting translation and/or RNA degradation [52].

The first ncRNA to be characterized was MicF (Figure 1) as a regulator of the outer membrane protein OmpF [34]. Since then, with the development and help of several techniques (which are described below), a plethora of new ncRNAs has been identified in several bacteria, mainly in *E. coli* and *Salmonella enterica* [53].

Broadly, ncRNAs can be divided into two major classes: (i) *cis*-acting ncRNAs that are transcribed from the opposing strand of their target mRNAs, with whom they present a perfect complementarity; (ii) *trans*-acting ncRNAs, which are encoded in a distinct location from their targets, thus they are only partially complementary with them, allowing the recognition of multiple targets by a single ncRNA [54]. In Gram-negative bacteria, the *trans*-acting ncRNAs often require the aid of a RNA chaperone, such as Hfq and/or ProQ, which act as matchmakers to promote attachment to the targets and to stabilize these RNAs (see section *RNA chaperones*). In addition, with the discovery of the CRISPR/Cas systems, a new class of ncRNAs has been considered (see also Section *The CRISPR System*).

Many bacterial ncRNA have been shown to have roles in virulence, and this has been observed both in Gram-negative and -positive bacteria (extensively reviewed in [55]). It was in the early nineties that the first ncRNA was shown to be implicated in pathogenesis—*S. aureus* RNAIII (Figure 1). This ncRNA is involved in the regulation of the *agr* quorum-sensing (QS) system, a key regulatory system that is engaged not only in virulence control [37], but also in antibiotic resistance mechanisms, autolysis, and biofilm formation [56,57,58,59]. Later on, additional ncRNAs were found to be involved in *S. aureus* pathogenicity [60].

**Figure 2 microorganisms-10-02303-f002:**
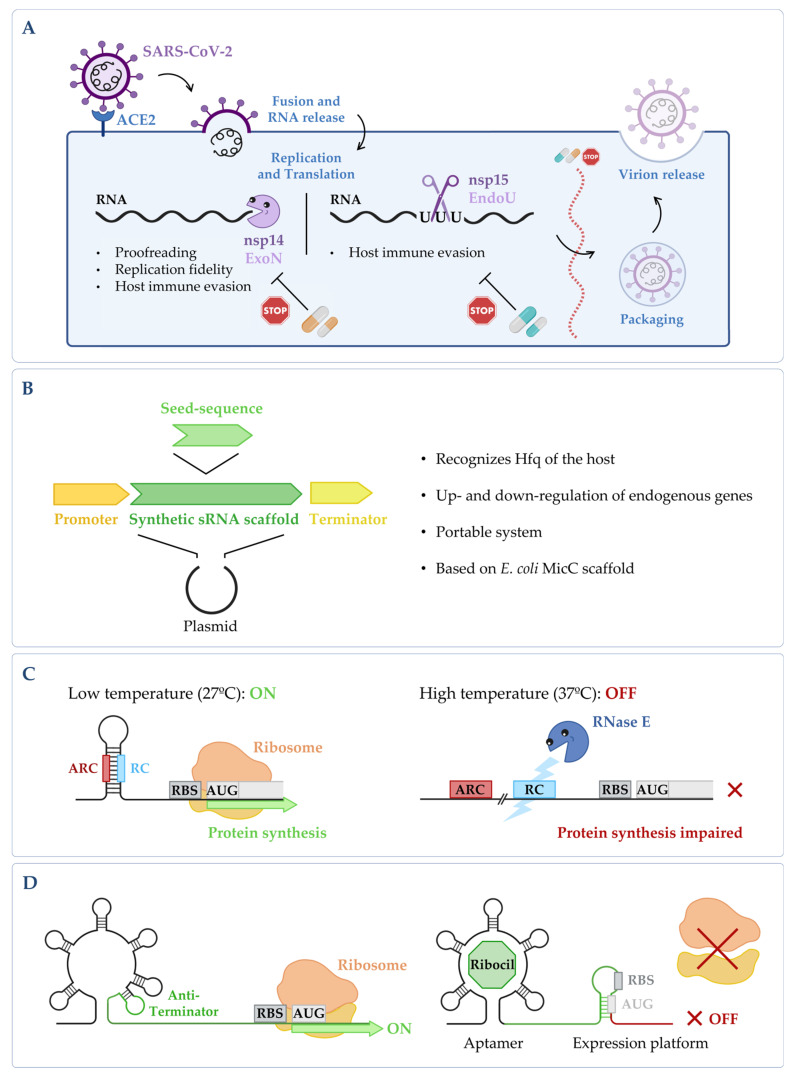
(**A**) *Targeting fundamental viral proteins to inhibit SARS-CoV-2.* The use of nsp14 and nsp15 ribonucleases as druggable targets may impair SARS-CoV-2 viral replication cycle, and therefore, it can be a good way to tackle infection. (**B**) *Synthetic sRNA expression system.* Vector containing the *E. coli* MicC scaffold, in which a customized seed sequence complementary to the endogenous target transcript is inserted [61]. (**C**) *RNase E mediated thermoregulation*. When temperature is low, the RNase E cleavage site (RC) is hybridized with the anti-RNase cleavage site (ARC) forming a hairpin, thus blocking the cleavage by RNase E and allowing gene expression to occur. When temperature increases, the RC is exposed, the mRNA is cleaved by RNase E, and the expression of the gene is impaired (RBS stands for ribosomal binding site and AUG for the initiation codon). This is adapted from [62]. (**D**) *Regulation of the FMN riboswitch by Ribocil*. FMN riboswitch in the absence of any compound; it presents a conformation that allows gene expression to occur. Upon binding to the riboswitch, Ribocil induces a rearrangement of its structure that sequesters the RBS, thus preventing translation; this is adapted from [63]. Figure created using BioRender.com (accessed on 11 November 2022).

*Salmonella* has been extensively used as model organism to study ncRNAs. Several ncRNA genes are encoded by the pathogenicity islands of a *Salmonella* virulent strain, with IsrJ playing a crucial role during the infection process [64]. Another example is the remarkably long *cis*-acting ncRNA, AmgR, which has been shown to affect *Salmonella* virulence [65]. In this foodborne bacterium, the OmpD is the most abundant porin, and therefore, its levels need to be tightly regulated to prevent cell lysis [66]. *ompD* mRNA was demonstrated to be regulated by the ncRNAs InvR, MicC, RybB and SdsR [67,68,69,70]. Interestingly, mutations on ncRNA genes that negatively regulate conserved outer membrane proteins (OMPs) in *Salmonella*, in combination with a mutation on a transcription regulator, allowed the rationale design of an attenuated vaccine for pigs [71]. Vaccination against *Salmonella* could improve animal health and reduce antibiotic usage, ultimately increasing food safety.

In the foodborne pathogen *Listeria monocytogenes*, several ncRNAs have been identified. Some of them are exclusively expressed inside macrophages or in human blood, suggesting an important role of these transcripts for intracellular growth [72,73,74,75]. Interestingly, it was described that two riboswitches (see section *RNA riboswitches*) could also act as conventional ncRNAs by interacting with the 5′ UTR of the *prfA* mRNA which encodes a master regulator for *Listeria* virulence [76].

In *Shigella,* it was demonstrated that RyhB and RnaG ncRNAs influence pathogenesis [77,78,79]. The Ssr54 ncRNA is important for the tolerance and virulence of *Shigella* under hyperosmotic pressure [80], and RyfA1, which is under the control of the ncRNA RyfB1 [81], impacts the levels of *ompC* mRNA that encode an OMP that is related with *Shigella* virulence.

In *Streptococcus pneumoniae*, a significant percentage from the identified ncRNAs has important roles in virulence traits [82]. srn157 and F32 were shown to be important for the adhesion/invasion of endothelial or nasopharyngeal cells [83,84], five redundant csRNAs (cis-dependent small RNAs) together with the ncRNA srn206 act to modulate competence [85], and srn135 was demonstrated to be involved in *pilus* regulation [86].

Native RNA-based interactions of ncRNAs with proteins, RNA transcripts and/or DNA are essential for coordinating gene expression. These interactions, due to their advantages when compared to conventional gene knockouts, are being increasingly targeted in synthetic biology, and in the antimicrobial and therapeutic fields. Engineering strategies involving *trans*-regulatory ncRNAs fall into two general approaches: (a) altering the expression of well-characterized natural ncRNAs to induce enhanced regulatory effects on protein levels, and (b) designing synthetic ncRNAs to knockdown the expression of individual proteins. The use of these synthetic ncRNAs as a gene-silencing tool mimics the RNA interference system (from eukaryotes) in bacteria. ncRNAs show high modularity, which enables synthetic biologists to decompose and recombine the ncRNA parts to engineer artificial riboregulators with different functions. The basic modular design includes a promoter, an antisense binding domain to target mRNA, a scaffold for stability and/or Hfq binding and a terminator. For instance, a MicC-based synthetic ncRNA was successfully used to fine-tune the gene expression in *E. coli* [87]. The plasmid-based synthetic ncRNA was easily transferred to other strain backgrounds. A similar construction with “tailor-made synthetic sRNAs” was developed in *P. putida* where the binding of native Hfq to a MicC scaffold was also demonstrated. It was shown that when these synthetic sRNAs were induced, they could actively control the gene expression [61]. The versatility of this system makes it very useful for different purposes (Figure 2B).

ncRNAs offer an additional suit of tools for engineering metabolic pathways in bacteria with an interest in industrial production, with several examples of successful applications. Despite being a less explored area, recent works also support promising applications in medicine. As mentioned above, the OMPs of Gram-negative bacteria play a main role in mediating bacterial antibiotic resistance and in the virulence of innumerous bacteria. Multiple natural ncRNAs have been found to control OMP expression [51]. Therefore, the use of synthetic sRNAs has been explored as a way to modulate OMP expression and modulate bacterial virulence [88]. There are also examples of synthetic ncRNAs that have been designed to control the virulence of pathogenic bacteria through the modulation of its cellular motility [88], antibiotic sensitivity [89,90] or to downregulate the expression of essential proteins that regulate mRNA turnover [91].

### 3.1. The Csr System

There are also ncRNAs that bind to proteins to alter their activity, but fewer examples are known in comparison to the antisense ncRNAs. The Csr (carbon storage regulator) system is very important for central metabolism (reviewed in [92]). It is composed of the CsrA (or RsmA) protein, an RNA-binding protein that modulates the expression of several mRNA molecules, and the ncRNAs CsrB and CsrC (or RsmY and RsmZ) that sequester the CsrA protein, thereby inhibiting its activity. A fourth component of this system is the CsrD protein that marks CsrB and CsrC ncRNAs to be degraded by RNase E [93,94,95,96].

In *Salmonella*, the deletion of *csrA* caused serious growth deficiencies and defects on invasion [97]. The deletion of both CsrB and CsrC significantly reduced the *Salmonella* Pathogenicity Island 1 (SPI1) gene expression, and, as a consequence, epithelial cell invasion [98,99]. Additionally, these two ncRNAs were shown to be required for the regulation of the type I fimbrial operon, which contributes to biofilm formation [100]. In *S. flexneri,* it was determined that CsrA activity is linked to virulence and to the cell membrane structure [101]. In *Legionella pneumophila*, it was demonstrated that the CsrA protein is crucial for replication inside the macrophages [102], affects the flagellar expression [102,103] and impacts the levels of important regulators of virulence-associated traits [104]. The other two components of the Csr system in *Legionella*, RsmY and RsmZ, were shown to be expressed depending on the growth phase, and the absence of both ncRNAs impaired the infection and interfered with the replication inside the host [105]. In *Vibrio cholerae* and in *Pseudomonas aeruginosa*, the Csr system is important for quorum-sensing regulation [106,107], and, in the plant pathogen *Erwinia carotovora* ssp. *carotovora,* it controls the production of extracellular enzymes and secondary metabolites [108].

### 3.2. RNA Chaperones

RNA chaperones facilitate the proper RNA folding, remodel the RNA structures to expose important regulatory elements, and, in several cases, they can protect the RNA molecules from degradation by ribonucleases. The two RNA chaperones that have been well described until now are Hfq and ProQ. For more information about their mechanism of action and their structural aspects, please see [109].

Hfq is a highly conserved protein from the Sm family, and it has homologues in approximately 50% of all of the sequenced bacteria [110]. This pleiotropic regulator was first described as an essential host factor of the RNA bacteriophage Qβ [111], but several studies have recognized its role in RNA metabolism and in bacterial pathogenesis.

In *Salmonella*, it was demonstrated that Hfq is important for virulence, considering its role in the motility, membrane composition, invasion, and expression of genes from the SPI1 [112]. In *L. pneumophila*, Hfq plays a role in the iron uptake and storage system, and mutants lacking this chaperone showed defects during growth and in pigmentation, being slightly less efficient in infecting amoeba and macrophages [113]. In the foodborne pathogen *L. monocytogenes*, Hfq was proven to be important for the tolerance to osmotic and ethanol stresses, for the long-term survival during amino acid starvation and in the pathogenicity in mice [114]. Similar observations were reported in other common pathogens, namely in *V. cholera* [115], *P. aeruginosa* [116], *Neisseria gonorrhoeae* [117] and *Francisella tularensis* [118], thereby implicating the Hfq protein in highly relevant pathogen-related mechanisms.

ProQ is a FinO-like protein that is specific to Gram-negative microorganisms. This protein was initially described as a factor that affects the activation of the osmoregulatory transporter ProP, and it has only recently been shown to be an important RNA chaperone that is involved in the regulation of ncRNAs [119]. Contrary to Hfq, ProQ interacts with many *cis*-acting ncRNAs, which means that they regulate a different series of genes [120]. The involvement of this protein in bacterial virulence has not been fully explored. However, it has already been seen that in *Salmonella*, the absence of ProQ causes a decrease in the expression of genes that are involved in the motility and chemotaxis pathways, leading to an impaired ability to infect HeLa cells [121]. In *L. pneumophilia,* two ProQ homologues were described (Lpp1663 and Lpp0148/RocC), and RocC was shown to be involved in natural competence [122]. Furthermore, in the plant pathogens *Dickeya dadantii* [123] and *Erwinia amylovora* [124], the loss of ProQ affected different processes, causing a decrease in the virulence rate. Although only a few pieces of evidence have linked ProQ with pathogenesis, we believe that this number will increase in the near future.

## 4. Regulatory 5′ Untranslated Region (UTR) Elements

Two classes of regulatory elements located at the 5′ UTR of mRNAs have been shown to play important roles in gene expression: RNA thermometers and RNA riboswitches. These RNA elements allow the bacteria to rapidly and efficiently react to environmental stimuli. Taking into consideration the mode of action and simplicity of these regulators, these RNA elements are very appealing for the development of new tools to regulate gene expression, and they can be used for many applications.

### 4.1. RNA Thermometers

RNA thermometers or thermosensors are molecules that sense temperature shifts, inducing a conformational change of the RNA molecule that will affect the expression of the downstream gene. Regulation by temperature using RNA molecules allows for a more rapid and cost-effective response of the cell. RNA thermometers act by regulating translational initiation: for instance, at lower temperatures, the Shine-Dalgarno (SD) region and/or the initiation codon are masked by a stable secondary structure; when the temperature increases, this region is melted, thus allowing translation to occur. For more information about the mechanism of action of these molecules, please read [125].

Due to their nature, RNA thermometers are not conserved, and this is a challenge for bioinformatic prediction [126]. The first RNA thermometer was discovered more than thirthy years ago, and it controls the development of phage λ by regulating the cIII protein [127]. Contrary to what was described for all of the other RNA thermometers that were later discovered, it allows translation to occur when the temperature decreases [127].

A temperature shift is one of the challenges that a pathogen faces during the infection process. As such, natural RNA thermometers are crucial regulatory elements that are involved in bacterial pathogenesis by controlling the expression of virulence genes. This is the case of the *agsA* gene in *Salmonella*, the *prfA* gene in *Listeria*, the *lcrF* and *ompA* genes in *Yersinia*, the *cssA* gene in *Neisseria*, the *toxT* gene in *Vibrio,* and the *ompA* and *shuA* genes in *Shigella* [128,129,130,131,132,133,134,135].

In recent years, a number of synthetic RNA thermometers was developed with success in diverse applications. Contrary to the natural molecules, they were designed to be simpler and more predictable to facilitate their usage [136]. Despite the fact that most of the synthetic RNA thermometers are heat inducible, recently we also assisted to the development of heat-repressible RNA thermometers that use an RNase E-mediated mechanism (Figure 2C) [62].

An interesting example of the use of these regulators in medicine is the development of RNA thermometers for microbial therapeutics in vivo. Two synthetic thermometers were designed to act between 32 °C and 46 °C, which could be used in three different in vivo scenarios to combat microbial infections: (i) the capacity to detect and respond to the host’s fever, (ii) the selective activation of the microbial function at a specific location using focused ultrasound (allowing a local delivery of the therapeutics), and (iii) the restriction of the survival of the administered microbes to the host’s body temperature and self-destruction at room temperatures, thus preventing possible environmental contaminations [137]. Considering that this study was performed with *E. coli*, further studies may be required to adjust this process to other pathogens.

### 4.2. RNA Riboswitches

Riboswitches are *cis*-acting RNA elements that recognize metabolites, thus modulating gene expression in response to specific small molecules. In bacteria, most of the riboswitches are located at the 5’ UTR of a particular transcript, and are composed by two functional domains: the ligand-sensing domain (or the aptamer domain) and the regulatory domain (or the expression platform). In certain conditions, a small molecule binds the aptamer domain, inducing a conformational change that stimulates the expression platform. The expression platform will act over the coding sequence, thus regulating its expression. For more information about the mechanism of action of the riboswitches, please read [138].

Riboswitches were first discovered in 2002 [139,140,141], and since then, they have been acknowledged as crucial contributors for the control of gene expression in many organisms. They can bind to a plethora of small molecules, from vitamins to sugars, amino acids or metals, and they can exert their function in different ways [142]. Additionally, in some pathogens, important genes related with virulence are controlled by riboswitches. This was demonstrated in *L. monocytogenes*, where the major regulator of virulence, PrfA, was shown to be controlled by two riboswitches, which also function as ncRNAs [76], and in *Clostridium difficile*, where it was shown that riboswitches are important for growth and infectivity [143].

The existence of riboswitches in pathogenic bacteria presents novel targets for drug development. This has led researchers to start to manipulate how riboswitches bind to their ligands in order to design new molecules that could be used as antimicrobials [144,145,146]. The high-resolution crystal structures of the riboswitches bound to their cognate ligands have helped to design potential inhibitors with improved drug-like properties [147,148,149,150,151,152,153]. From the developed compounds, Ribocil (Figure 2D), which is currently in preclinical development, was shown to inhibit the growth of different bacterial strains, including methicillin-resistant *S. aureus* and *Enterococcus faecalis* [63,154,155].

Natural riboswitches combine both the sensory and regulatory functions. This principle of direct RNA-ligand interaction was exploited to synthetically design the aptamer-based conditional gene expression systems. Aptamers are single-stranded RNA or DNA molecules that can self-fold in a unique 3D-spatial conformation to specifically interact with their targets. The selection of aptamers with the capability to bind a plethora of different ligands can be performed in vitro through the so-called Systematic Evolution of Ligands by Exponential Enrichment (SELEX) Technology [156,157,158]. Aptamers targeting pathogenic bacteria and viruses have attracted increasing attention [159]. Such aptamers can be used for the specific recognition of infectious agents or to block their functions [160,161].

## 5. The CRISPR System

Upon a viral or plasmid invasion, bacteria (and archaea) integrate short fragments of foreign DNA into the host chromosome, namely, at a (variable) number of short repetitive *loci* (approximately 20–50 base pairs) known as the CRISPR, in a stage called adaptation. These exogenous DNA fragments are inserted by the Cas proteins, Cas 1 and Cas 2, which are the only Cas proteins that are conserved amongst all the CRISPR–Cas systems. The repetitive *loci* are subsequently transcribed and processed into a library of short CRISPR-derived RNAs (crRNAs) that are complementary to the previous invading nucleic acids. This is the stage of crRNA expression and biogenesis. Then, comes the interference stage, in which each crRNA can guide the effector nucleases to destroy the foreign genetic material through specific cleavage [162,163]. Thus, the integration of invasive DNA constitutes a genetic record of prior encounters with the transgressors, and reflects the surrounding environmental conditions, which change over time.

The CRISPR systems can be divided into two main classes, class 1 and class 2. The class 1 system is found in 90% of the CRISPR *loci* in bacteria and archaea, whereas the class 2 systems only represent 10% of the CRISPR *loci* that are found in bacteria. The specific types within each class are defined by the effector endonuclease— the Cas protein—which is responsible for cleavage [164,165].

The Cas effector proteins are, thus, non-specific nucleases that can be programmed by small guide RNAs, the crRNAs, to be directed to target DNAs or RNAs. Great emphasis has been given to these systems due to these RNA-guided programmable enzymes which exhibit remarkable flexibility in targeting. These have encouraged an ever-expanding array of applications. The most explored and used toolbox in genomic engineering is the class 2 (type II) system, which is better known as CRISPR–Cas9. Cas9 is the characteristic effector protein, and it is essential for immune mechanisms in bacteria [166]. Furthermore, CRISPR–Cas9 are also abundant in pathogenic and commensal bacteria. Indeed, the *cas9* gene has been reported to play an important role in controlling virulence in various pathogens [166,167,168]. As a virulence regulator, Cas9 is involved in specific steps of the pathogenesis of different bacterial species, as well as in common processes of virulence.

In *Streptococcus* sp., Cas9 was reported to influence key regulators of virulence traits, such as adhesion and infection [169,170]. The same effect was verified in the knockout strains lacking Cas9 in *N. meningitidis* [171]. Curiously, *cas9* deletion in *Campylobacter jejuni* highly affects its sensitivity to antibiotics, regulating several genes that promote antimicrobial resistance [172]. This proves the connection of CRISPR with antibiotic resistance mechanisms. Interestingly, in the case of *L. pneumophila*, Cas2 and not Cas9 is the CRISPR enzyme that is involved in the infection process of macrophages [173]. Both the Cas9 and Cas2 proteins belong to the same CRISPR–Cas type II system. Although they maintain conserved functions regarding their role in the CRISPR bacterial immunity, they appear to have different functions in virulence, depending on the microorganism.

The interest in the relationship between CRISPR and virulence has grown, and it was later discovered that the CRISPR–Cas type I systems also have an important role in the evasion of bacteria from the host. *Streptococcus mutans* contains a class 1-type I CRISPR, whose effector protein is Cas3. In the absence of the *cas3* gene, the strain formed less biofilm, became more sensitive to fluoride, and the expression of the virulence genes was significantly downregulated [174]. Similar observations have been reported with the *S. enterica* isolate 211 [175]. Additionally, in *P. aeruginosa* UCBPP-PA14, the *cas3* gene has been shown to be involved in the achievement of lower pro-inflammatory host responses in cell and mouse models [176].

Biofilm development and antibiotic resistance are intimately connected since the biofilm matrix can delay the penetration of antimicrobial agents. Biofilm formation is a highly regulated process, and CRISPR has proven to be one of these regulators. Most pathogens involved in nosocomial infections have biofilm-forming abilities. Interestingly, an increased ability to form biofilms has been reported in CRISPR–Cas positive *Enterococcus faecalis* and *P. aeruginosa* strains [177]. Additionally, in *Acinetobacter baumannii*, specific genes that are involved in biofilm formation appear almost exclusively in strains that are enriched with CRISPR–Cas systems [178]. It also appears that CRISPR contributes to a tight control depending on the surrounding environment. The lysogenic infection of *P. aeruginosa* UCBPP-PA14 by the bacteriophage DMS3 inhibits biofilm formation and swarm motility in a manner that is dependent on the CRISPR regions and *cas* genes [179]. This strategy, by preventing the infected bacteria from forming biofilms and performing other group behaviors, can limit the effects of bacteriophage spread in bacterial communities.

The existence of group behaviors among the bacteria is indeed extremely important. During biofilm formation, bacteria have the ability to communicate with each other through the process of QS. In *Serratia marcescens*, it appears that CRISPR–Cas immunity is integrated into the QS circuit, enabling greater defense at higher cell densities [180]. Similarly, *P. aeruginosa* UCBPP-PA14 also uses the QS process to activate *cas* gene expression [181]. Thus, bacteria seem to be able to use QS communication to control CRISPR–Cas expression according to the needs of the cell.

In 2011, Charpentier and co-workers [182] reported the existence of a *trans*-encoded small RNA (tracrRNA) that was transcribed upstream and in the opposite strand of the CRISPR *locus*, with 24 nucleotides that were complementary to the repeat regions of the crRNA precursor transcripts (pre-crRNA). This tracrRNA is responsible for pre-crRNA maturation by promoting the cleavage of the tracrRNA-pre-crRNA duplex by the very well-known and widely conserved endoribonuclease RNase III [182]. Soon after this, tracrRNA was reported to trigger Cas9 to cleave the target DNA [41]. This discovery enabled the development of a breakthrough method of genome editing, which was later recognized by being awarded the Nobel Prize in Chemistry in 2020 to the scientists, Emmanuelle Charpentier and Jeniffer A. Doudna [41] (Figure 1). There is already evidence that ncRNAs related to the CRISPR systems play a role in bacterial virulence. In *Francisella novicida*, Cas9 uses a small CRISPR/Cas-associated RNA (scaRNA) to repress an endogenous mRNA transcript encoding a bacterial lipoprotein, which elicits a proinflammatory innate immune response in the host [168]. A CRISPR-associated ncRNA, RliB, has also been shown to play a role in *L. monocytogenes* pathogenesis [72]. Thus, it appears that ncRNAs constitute an extra layer in CRISPR regulation.

The knowledge of CRISPR has opened avenues to the entire scientific community for the development of genetic engineering tools, namely in the creation of new and improved versions of CRISPR systems that are revolutionizing the world today. As the pieces of the CRISPR puzzle are being discovered, more and more applications are emerging. For instance, in 2014, the use of a type I CRISPR–Cas system in *E. coli* enabled the successful removal of individual bacterial strains from mixed populations, which share a high homology. This highlights the extraordinary specificity of this tool, and has opened up the possibility of developing smart antibiotics that prevent multidrug resistance and differentiate between the pathogenic and beneficial bacteria [183].

These novel antibacterial strategies can be based on CRISPR–Cas systems, primarily on CRISPR–Cas3 and CRISPR–Cas9, to target DNA, which can be designed to specifically eliminate the plasmids that carry antibiotic resistance genes and chromosomal virulence genes, among others, in order to attack the pathogens (Figure 3A). The tool consists of integrating the CRISPR–Cas sequences into a plasmid vector, allowing the system to target and cut genes of interest. A system that was identified more recently by Feng Zhang’s lab, CRISPR–Cas13 (class 2), brought a new perspective to the CRISPR tool. The RNase Cas13 cleaves single-stranded RNA (ssRNA) molecules in a crRNA-guided manner [184]. CRISPR–Cas13 also exhibits the promiscuous degradation of ssRNAs when it is performing targetted RNA cleavage, thus, limiting the host cell growth by inducing dormancy in the bacteria [185]. Additionally, unlike Cas9-based antimicrobials, the CRISPR–Cas13 system exhibits strong bacterial killing activity, regardless of the target genes’ location (chromosome or plasmid) [186] (Figure 3A). This system has been successfully tested by constructing antibacterial nucleocapsids (CapsidCas13) that are capable of killing carbapenem-resistant *E. coli* and methicillin-resistant *S. aureus* through the recognition of the corresponding antimicrobial resistance genes [186].

Nevertheless, these CRISPR–Cas tools are still limited in terms of their clinical application due to their delivery systems. The use of conjugative plasmids [188], phage vectors [189,190], membrane vesicles [191] or their encapsulation into nanomaterials [192] have been explored as delivery systems.

CRISPR has also received substantial attention as a diagnostic tool due to its potential to detect nucleic acids in a quick, sensitive and specific manner [187] (example in Figure 3B). Within the current pandemic context, CRISPR diagnostic technologies were quickly adapted and optimized [193,194,195], being recently highlighted as one of the seven technologies to watch in 2022 [196] (Figure 1).

The role of CRISPR–Cas systems in modulating the genotypes, physiology and ecology of bacteria, plus the implication of CRISPR–Cas in limiting horizontal gene transfer, or in enabling the acquisition of advantageous genes are topics of great interest, as is the development of CRISPR for new applications in the area of treatment of infectious diseases. However, the application of CRISPR–Cas antimicrobials remains at a very preliminary stage and numerous obstacles await to be resolved.

## 6. RNA Technology

### 6.1. RNA Sequencing (RNA-Seq)

The development of Next Generation Sequencing (NGS), which is also referred to as deep-sequencing, or high-throughput sequencing, has provided a set of diverse modern technologies with applicability to the study of DNA, RNA and proteins [197]. In particular, RNA-seq methodologies allow for the determination of the sequence of an overwhelming amount of different RNA molecules in a massively parallel way [197,198].

Nowadays, there is a panoply of distinct RNA-seq-based approaches that aim to uncover and characterize the RNA species being expressed at each moment in a cell culture or a single cell. Many fields of study have benefited from such methodologies [199,200,201,202]. In microbiology, RNA-seq derived technologies have been useful as tools for various purposes such as the optimization of bacterial chassis for industrial biotechnology [203] and synthetic biology [61,204], and for the study of both human microbiota [205,206] and human pathogens. In this section, we will present some examples of the contribution of different RNA-seq protocols for the study of pathogenic microorganisms (reviewed in [201,202]).

In a recent study, messenger RNA sequencing (mRNA-seq) was used to elucidate the function of a specific gene which was postulated to be involved in the virulence of the zoonotic bacterial pathogen *Streptococcus suis* type 2 [207]. In another study, Quant-seq, a variation of mRNA-seq, which is more focused on the 3′-end sequences of polyadenylated RNAs [208], served to demonstrate that human neural progenitor cells infected by Coxsackievirus B3 change their expression patterns, upregulating antiviral innate immunity and inflammatory pathways during infection [209].

As already addressed, ncRNAs are crucial regulators. NGS, and particularly small RNA sequencing (sRNA-seq), has largely contributed for the identification of new ncRNAs species in several pathogenic microorganisms [72,74,82,210].

Moreover, RNA modifications can influence the structure, stability, decoding, and recognition of RNA molecules. They often occur during transcription (e.g., the 5′ NAD cap) or post-transcriptionally (e.g., methylation resulting in N^6^-methyladenosine, m^6^A), and they may also play a prominent role both in the bacterial stress response and pathophysiology, and in host adaptation [211,212]. Combining mass spectrometry (MS) with RNA-seq procedures allows for the precise localization of the RNA modifications and the study of their dynamics [213]. Remarkably, specific RNA-seq methodologies have been applied to bacterial pathogens to detect the RNA modifications that are crucial for cytotoxicity and virulence, such as NAD capture-seq which measures the NAD incorporation [214,215], and m^6^A-seq which identifies the methylated residues in the transcripts [216].

In biology, understanding the network of interactions in the cell is crucial. The RNA interaction by ligation and sequencing (RIL-seq) was designed to identify the RNA–RNA interactions, and this has been particularly useful to elucidate pairs of ncRNAs and their respective mRNA targets. In pathogenic *E. coli*, this technique was sufficient to determine the global interactome of RNA molecules binding to Hfq, further detecting ncRNAs that had not been previously annotated [217]. In turn, gradient profiling by sequencing (Grad-seq) was developed to analyze the native RNA–protein complexomes in the cellular environment. It combines two approaches (RNA-seq and liquid chromatography-tandem mass spectrometry (LC-MS/MS)) [119], and it can: identify major RNA–protein complexes and RNA binding proteins, cluster ncRNAs according to their biochemical properties, and complement the information regarding the function of domains of uncharacterized proteins [218]. In fact, thanks to this technology, the ProQ was discovered as an important RNA chaperone, which was a missing piece in the puzzle of ncRNA regulation [119].

One of the major breakthroughs in this field was the establishment of single-cell RNA sequencing (scRNA-seq), which allows for the discrimination between RNA species being expressed in different cells belonging to the same population or different populations in the same sample (reviewed in [219]). This technology gained special relevance in enlightening the mechanism of infectious diseases in several pathogens [220,221,222]. Currently, a promising trend in the scRNA-seq approaches is the incorporation of droplet- and microwell-based microfluidics, improving sequencing throughput in an affordable, portable and scalable way [223].

In the last decade, differential RNA-seq emerged with the advantage of distinguishing between the primary and processed transcripts. This way, it has provided an opportunity to map the transcriptional start sites (TSS), and exposed the existence of pervasive transcription and a generally high abundance of ncRNAs in the bacterial genomes [224,225,226,227].

Differential RNA-seq served as an inspiration for dual RNA-seq which has the capability of sequencing RNA molecules of two or more species simultaneously [42] (Figure 4A). The main goal is to get the best possible approximation to the in vivo conditions (reviewed in [201,202]). Although there are still many limitations to overcome, the dual RNA-seq advantages are undeniably evident: it brings the possibility of directly evaluating which genes are differentially expressed in each interacting species which can then be mapped against the known interaction networks or used to predict novel gene regulatory networks [228]. This tool has been very important for unravelling the mechanisms of infection of several pathogens [229,230,231,232,233] (Figure 1). A surprising example of triple RNA-seq enclosed RNA isolation and sequencing starting from a sample containing human immune cells, *Aspergillus fumigatus* (fungus) and Cytomegalovirus (CMV) [234].

While NGS technologies usually produce short reads, the Third-Generation Sequencing (TGS) has emerged, enabling the sequencing of longer fragments (long reads). As the raw reads can be disclosed in real time, TGS permits data interpretation to occur prior to the samples being fully sequenced [235]. There are two main TGS categories: single-molecule real-time (SMRT) sequencing, and nanopore single-molecule sequencing (Figure 4B). Distinctively, nanopore sequencing relies on registering the changes in the electrical current during the translocation of the template molecule along a protein nanopore, rather than recording the optical or chemical signals that are emitted during the polymerization of a complementary strand, as it commonly happens in other RNA-seq techniques [236]. In the cases where this technology directly uses an RNA molecule as template it may then be called direct RNA-seq. These TGS methods have been of particular relevance for studying pathogenic microorganisms to further disclose the link between post-transcriptional RNA modifications and microorganisms’ mutability and virulence [237,238], as well as to characterize transcript isoforms [239,240,241].

Overall, when they are compared with first-generation sequencing (Sanger sequencing), the NGS and TGS methods are faster, more sensitive and produce a greater amount of data encompassing a wide repertoire of RNA molecules [236]. The employment of NGS and TGS in the meta-transcriptomics through whole-genome or full-length 16S rRNA sequencing has already been shown to accelerate the diagnosis of infectious diseases, namely, by reducing the waiting time, improving the pathogen taxonomic classification and the effectiveness in the detection of RNA viruses, and by extending the spectrum of antibiotic resistance genes that are detected in clinical samples [242,243].

Finally, the above-mentioned RNA-seq strategies might help in the identification of diagnostic biomarkers, the choice of the appropriate treatment for different severity stages of a certain disease, of drug target candidates and potential drugs which can also be repurposed and used for the efficient treatment of specific infectious diseases [244,245].

In fact, independently of the specific RNA-seq method that is employed, it will always require bioinformatic pipelines to process the enormous volume of data. In the past, programming skills were a prerequisite, but many tools with graphical user-friendly interfaces have been progressively developed and made accessible for everyone, as it is the case of the Galaxy platform [246]. Many online resources are also available, namely, several specific transcriptome browsers, or simply, brief explanations of the different techniques, protocols and data.

**Figure 4 microorganisms-10-02303-f004:**
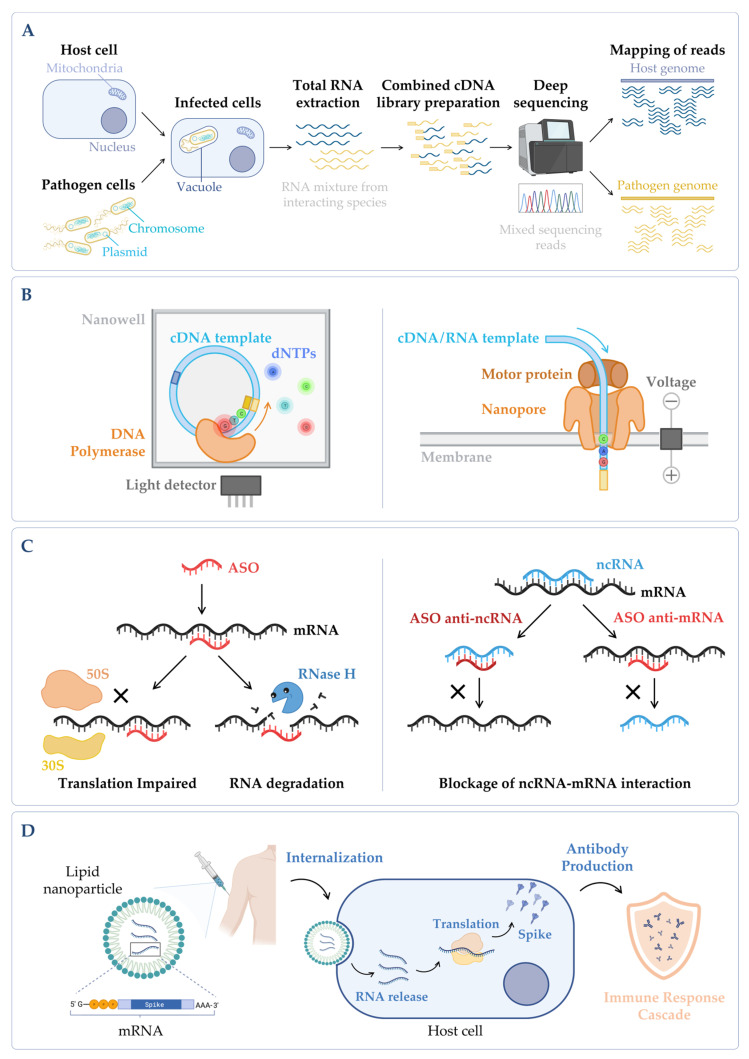
(**A**) *Simplified workflow of a dual RNA-seq protocol*. Host cells are infected in vitro with pathogen cells, lysed and total RNA is extracted. The sequencing library is prepared, and sequencing is performed in a NGS platform, obtaining simultaneously the results for both species. During bioinformatic data analysis, after quality control and data cleaning, the reads from the host and the pathogen are separated in silico in the mapping step. Annotation and quantification are carried out independently for each species, allowing to analyze host and pathogen differential gene expression in parallel, as well as to predict functional correlations between species [230]. (**B**) *Main categories of third-generation sequencing (TGS)*. (Left panel) Single-molecule real-time (SMRT) sequencing—Sequence is determined through emission of fluorescence due to the incorporation of a fluorescently labelled deoxyribonucleotide (dNTP) by the DNA polymerase in the nascent complementary strand of the cDNA template molecule. The DNA polymerase is anchored to the bottom of a nanowell. (Right panel) Nanopore sequencing—Sequence is obtained without imaging. The template nucleic acid is bound to a motor protein which takes the molecule to a protein nanopore. When the template molecule is translocated through the pore, each nucleotide with its own modifications produces a characteristic current shift that is recorded. Unlike the other methods, direct RNA-seq uses an RNA molecule as template [236]. (**C**) *Antisense oligonucleotides (ASOs) mechanism.* (Left panel) General mechanism of ASOs activity. The oligonucleotide binds to the complementary RNA, impairing ribosome progression and/or causing transcript cleavage of a target duplex of mRNA/ASO by RNase H. (Right panel) Targeting of ncRNA–mRNA interaction. In this case, the ASO can be designed to mimic the ncRNA and block its binding to the mRNA (anti-mRNA ASO) or mimic the mRNA sequence to sequester the ncRNA (anti-ncRNA ASO) [247]. (**D**) *mRNA vaccines mechanism*. The nucleoside-modified mRNA containing the coding sequence of the protein of interest (SARS-CoV-2 Spike protein) is encapsulated in a lipid nanoparticle (LNP). Upon human vaccination, the LNP is internalized, and the mRNA coding sequence is recognized by the host translation machinery, leading to the production of Spike proteins. This will induce the production of specific antibodies by the host immune system, inducing an immune response cascade [248]. Figure created using BioRender.com (accessed on 11 November 2022).

### 6.2. ASOS—The Use of Antisense Antimicrobial Therapeutics

An alternative strategy to fight the growing antibiotic resistance phenomena is to design gene-specific oligomers that can specifically target any single pathogen. Antisense antimicrobial therapeutics are a biotechnological form of antibiotic therapy using short, single-stranded oligomers that mimic the structure of DNA or RNA and bind to specific, complementary RNA in a target organism [249,250]. In microorganisms, ASOs (antisense oligonucleotides) bind to their complementary mRNA and inhibit its translation into proteins through the steric blockage of the ribosome progression and/or by promoting the degradation of the targeted mRNA through the RNase degradation of the ASO/mRNA duplex [250].

A key advantage of this antisense approach is that ASOs can be rationally designed to target any microbe through sequence complementation, thus, significantly enlarging the available selection of potential therapeutic targets [249]. A main goal in ASO design is the achievement of high specificity with minimal off-target effects. The sequence specificity and the short length of the antisense antimicrobials pose a minimal risk to human gene expression. Moreover, the specificity of antisense antimicrobials avoids the non-selective killing of the beneficial commensal bacteria by broad-spectrum antibiotics. This overcomes the unintended side-effects that are caused by the dysbiosis of the microbiome, and the consequent medical complications.

The use of antisense therapeutics has been progressively advancing towards clinical use, but in recent years the field has been accelerating. The identification of essential genes and the number of sequenced genomes has largely contributed to this. However, despite the fast advances in the eukaryotic fields [251], the progress in the use of ASOs as antibacterials has been delayed due to the poor uptake efficiency of the antisense molecules by bacteria [249]. This is mainly due to the electrostatic charge or the size barrier that is imposed by the cellular envelope (plasma membrane and cell wall). Other challenges regarding ASO efficiency are its intracellular concentration, oligomer length, nuclease resistance and binding kinetics.

ASOs are typically 10–30 nucleotides in length. The cellular nucleases rapidly attack the unmodified ASOs. Therefore, numerous chemical modifications have been described (e.g., phosphorothioates, locked nucleic acids, peptide nucleic acids, and phosphorodiamidate morpholino oligomers) to confer resistance against nucleases, to improve the stability of the ASO/mRNA hybrid formation and/or to preserve the target specificity.

In the sense of overcoming the challenge of bacterial cellular uptake, the most common strategy for facilitating antisense oligonucleotide delivery is the conjugation of a cell-penetrating peptide (CPP) to the antisense oligonucleotide. The attachment of a compound that can penetrate the bacterial cell wall facilitates the delivery of synthetic antisense oligomers into the bacterial cytoplasm. CPPs are short cationic or amphipathic peptides, which are usually composed of less than 30 amino acids. CPPs have been used with success to deliver modified ASOs in different bacteria ([252] for a review).

Phosphorodiamidate morpholino oligomers (PMOs) are synthetic single-stranded oligomers with a modified backbone which makes them resistant to nucleases [250]. The use of CPP-PMOs has been effective against infections caused by antibiotic resistant bacteria of the genus *Acinetobacter* (*A. lwoffii* and *A. baumannii*) and *Klebsiella pneumoniae* [253]. Wesolowski et al. described a CPP-PMO conjugate that targeted *E. coli gyrA*, a highly conserved gene that is found across multiple bacterial species [254]. The authors show that *gyrA* CPP-PMO reduced the viability of both the Gram-positive and Gram-negative bacterial strains (*Enterococcus faecalis*, *Staphylococcus aureus*).

GyrA mRNA was also targeted in *S. pyogenes,* but it used a CPP-PNA. Peptide nucleic acids (PNAs) are constructed by attaching bases to a modified polyamide backbone. The PNAs are uncharged, which in part accounts for their high affinity for RNA [255]. Successful examples of PNA targeting in different bacteria have been described [256,257]. In the foodborne pathogen *C. jejuni*, the *cmeABC* operon encodes a multidrug efflux pump that confers resistance to a broad range of antibiotics [258]. The use of PNAs targeted to different regions of the *cmeABC* operon restored the antibiotic susceptibility [259].

Locked nucleic acids (LNAs) are oxyphosphorothioate analogues with a 2′-O,4′-C-methylene bridge that locks the ribose ring in the C3′-endo conformation [260]. Both the CPP-PNAs and CPP-LNAs have been used in *S. aureus* to target the *ftsZ* mRNA, a gene that is required for cell division [261,262].

As it is mentioned in the previous sections, RNase E is an essential enzyme that is highly conserved in Gram-negative bacteria, and it has no known human orthologue [2]. Thus, the *rne* gene is a good target for antisense antibiotic development. Using *E. coli* as a model, Goddard and colleagues have used LNA gapmers, oligonucleotides consisting of a central region of DNA that is flanked by regions of chemically modified LNA nucleotides, to target RNase E [263]. Using this antisense antibiotic strategy, the authors were able to block the translation activity and trigger the RNase H-mediated cleavage of the *rne* mRNA in vitro, introducing the way to the use of this novel anti-bacterial target in different pathogens (Figure 4C, left panel).

Beyond the targeting of essential genes to reduce the viability of the pathogens, an alternative strategy for using antisense antibiotics is to target non-essential genes, which are required for virulence. Some examples of these are the genes required for invasiveness, biofilm formation [264], and antibiotic resistance genes. In this latter case, the co-administration of the PMO with the antibiotic would restore the susceptibility of the bacteria to its administration [264].

There are also other levels through which ASOs can reprogram the gene expression. For instance, ASOs can target the regulation by ncRNAs over their mRNA targets. In this case, the ASO can be designed to mimic the ncRNA and block its binding to the mRNA target (anti-mRNA ASO) or mimic the mRNA sequence to sequester the ncRNA (anti-ncRNA ASO). In both cases, the ncRNA–mRNA interaction is impaired (Figure 4C, right panel).

Henderson and co-workers [247] designed PNAs to target the ncRNA–mRNA interactions related to a QS system in *V. cholerae*. The Qrr ncRNAs are composed of four redundant regulators that target, among other genes, the *hapR* mRNA. At a low cell density, the expression of Qrr ncRNAs represses the master regulator HapR to promote the host colonization and virulence factor production in this human pathogen. At a high cell density, attained at later stages of the infection, the Qrr ncRNAs are no longer expressed, thus reactivating HapR expression and causing the release of the bacterium from the host. The use of two CPP-PNAs designed to sequester the Qrr ncRNAs (anti-Qrr ncRNA ASOs) prevented the Qrr-*hapR* mRNA interaction. This impaired the HapR downregulation, locking it in the HapR expression state (high cell density profile), with antibacterial implications.

The specific inhibition of a riboswitch by an ASO lead to the inhibition of the growth of *S. aureus, L. monocytogenes* and *E. coli*, which widen the lists of possible targets of this antimicrobial alternative system [265]. The potential of the applications of the different types of chemically modified ASOs and the creation of new and improved carrier compounds will expand their uses in multiple pathogenic bacteria.

### 6.3. RNA Interference (RNAi)

RNA interference (RNAi) is a biological process in which small ncRNAs recognize a specific mRNA, thereby promoting their degradation by Argonaute proteins, thus leading to gene silencing. This eukaryotic mechanism works as an innate defense mechanism against invading viruses [266]. The RNAi system was first described in 1998, and its important role in gene regulation rendered a Nobel Prize to Andrew Fire and Craig Mello [43] (Figure 1). Soon after their discovery, small interfering RNAs (siRNAs) were explored as a tool to treat several diseases, including viral infections [267,268]. The use of siRNA as a therapeutic agent implies the delivery of these molecules into the target cells, thereby activating the RNAi mechanisms in order to silence a specific gene. siRNAs have a high degree of specificity, targeting a unique mRNA, they have reduced toxicity and can reach inaccessible targets. The use of siRNAs to inhibit the replication of SARS-CoV [269], SARS-CoV-2 [270], respiratory syncytial virus (RSV) [271], and hepatitis C virus [272] has already been demonstrated, and this validates the potential of these molecules for the treatment of viral infections. The siRNA molecules target regions of the viral genome that are important for replication, such as mRNA that codes for the spike protein from SARS-CoV-2 [270], or the mRNA that codes for the nucleocapsid protein from RSV [271]. There are still some limitations for the use of siRNA-based therapies, such as siRNA stability, effective carriers, delivery routes and off-target effects. Regardless, clinical trials have already been performed with siRNA-based drugs to treat Ebola and RSV infections ([273], and reviewed in [274]).

### 6.4. mRNA Vaccines

Vaccination continues to be the most successful and cost-effective public health intervention to control and prevent infectious disease outbreaks. In fact, the conventional application of inactivated, live-attenuated or subunit vaccines had enormous success in the eradication of several infectious diseases, with a classic example being the complete eradication of the smallpox virus; however, many others were not as efficient in treating human immunodeficiency virus (HIV), *M. tuberculosis* and *Plasmodium* spp. [275] and other common vaccine-preventable diseases such as influenza [276].

Despite the promising results in the mRNA therapeutics field [36,277], mRNA was seen as too unstable and expensive to be used as a drug or a vaccine for several years [278]. A landmark experience was performed by Robert Malone in 1989 when he discovered the possibility of transfecting mRNA into eukaryotic cells which would induce their intracellular translation, thus recognizing the potential of exploring the RNA molecule for therapeutic purposes [279]. A year after this, the same principle was successfully applied in vivo [36]. In the 1990s, mRNA was tested as a therapeutic agent for the first time using lipid nanoparticles (LPNs) as the delivery method [277,280,281] (Figure 1). At that point, the challenges were to overcome RNA instability, to control the excessive host inflammatory responses and also to improve in vivo delivery systems. Katalin Karikó and Drew Weissman were central players in this context. They unraveled that the incorporation of modified, naturally occurring nucleosides in the mRNA molecules, particularly pseudouridine, prevents the activation of the immune response, reducing the synthetic mRNA immunogenicity in vivo [282] and provides a higher translation capacity [38,283] (Figure 1). More recently, it was demonstrated that N^1^-methylpseudouridine could provide even better results [284]. In addition, LNPs have become one of the most appealing and commonly used mRNA delivery tools [285].

In face of a sudden new coronavirus pandemic, previous advances in mRNA technology have enabled the rapid release of two highly efficacious mRNA vaccines in the market, BNT162B2 by Pfizer-BioNTech [248] and mRNA-1273 by Moderna [286]. Both of them are LNP-formulated nucleoside-modified RNA vaccines that encode the spike protein of SARS-CoV-2 as the target antigen (Figure 4D). These were the first mRNA-based vaccines to gather an emergency FDA approval, and their success in providing a robust immune response against SARS-CoV-2 was a game changing in the world of immunology and vaccine development (Figure 1). An important point to make is that the speed at which the COVID-19 vaccines were developed was influenced by a global emergency that resulted in an unseen alliance of the scientific community and in a massive funding.

For HIV, since the virus was reported in 1981, many unsuccessful attempts to produce a vaccine were announced [275,287,288,289], but Moderna has currently two mRNA vaccine candidates which are in Phase 1 clinical studies [290]; for tuberculosis and Malaria, the BioNTech company has announced that it is planning to move forward with the clinical trials of mRNA formulations for both of the diseases; for the influenza virus, tremendous effort has been invested in improving the current vaccines, and it is believed that the mRNA platform is well positioned to address the significant unmet need in the season flu [291,292,293]. Finally, the application of mRNA-based therapeutics is also being evaluated for other priority diseases by the CureVac and Moderna companies, such as Rabis, Respiratory Syncytial Virus, Human Cytomegalovirus, Human metapneumovirus and parainfluenza virus, Zika virus, Epstein–Barr Virus, Nipah virus and Chikungunya Virus. For the Ebola Virus and for *Streptococcus* sp. infections, preliminary mRNA vaccine studies in animal models are already being developed [294,295].

With increased scientific interest in this area, the next-generation mRNA technology will continue to mature both for vaccine development and therapeutics. The field of nucleotide-based vaccines came to the spotlight as a novel, faster and cheaper way to achieve vaccine development when compared with the conventional technologies [296]. Nevertheless, improvements in the storage and stability, production costs, geographic distribution capacity and research alliances are essential to ensure a more effective and prompt response to fight current and future endemic and/or pandemic infectious diseases.

## 7. Conclusions

RNA is back in the spotlight. The diverse role of RNA in all biological processes, together with the recognition of its important functional properties, have led to its exploitation in a wide range of biotechnological and medical applications. A great contributor to this change of perspective was the validation of the mRNA vaccines at an unpredictable scale and speed at which they fought against the COVID-19 pandemic. As they are natural molecules, RNAs present, in general, low toxicity and immunogenicity. The use of RNA elements presents advantages such as its independent control, tunability, composability and portability which empower their use as genetic tools. However, the advances in the application of these RNA tools have been limited by the rhythm of the progression of the technological advances which have enabled the characterization of new molecules and biological mechanisms. Built upon decades of scientific research, robust and prompt RNA technologies have now emerged, highlighting the importance of fundamental and applied research. For instance, we have testified in recent years, a fast discovery for the new classes of RNA molecules and molecular mechanisms that have transformed our comprehension of RNA metabolism. This review puts together the major discoveries regarding the connection between RNA metabolism and pathogenesis, and how this knowledge has been used to create new strategies to fight microbial pathogenicity. Antibiotic resistance is a serious problem that requires the creation of alternative therapeutics. As such, several RNA tools have surfaced as alternatives to control the virulence of pathogenic bacteria, namely, using synthetic non-coding RNAs, antisense antimicrobial therapeutics with antisense oligonucleotides (ASOs) or CRISPR–Cas antimicrobials. The application of these tools in prokaryotic organisms has been limited by different obstacles. In the case of ASOs, their use has been mostly limited by the development of delivery systems to improve their uptake by the bacterial cells. The same has happened with the CRISPR–Cas tools, and despite the new delivery systems which have been used with success, more research is needed to assure their safety and effectiveness. In the case of the mRNA vaccines, their implementation was possible thanks to the curiosity-driven studies of lipids and experiments with synthetic mRNA. The establishment of mRNA vaccines seems promising because of the speed with which they can be developed and produced, and their flexibility and adaptability to variants.

## Figures and Tables

**Figure 3 microorganisms-10-02303-f003:**
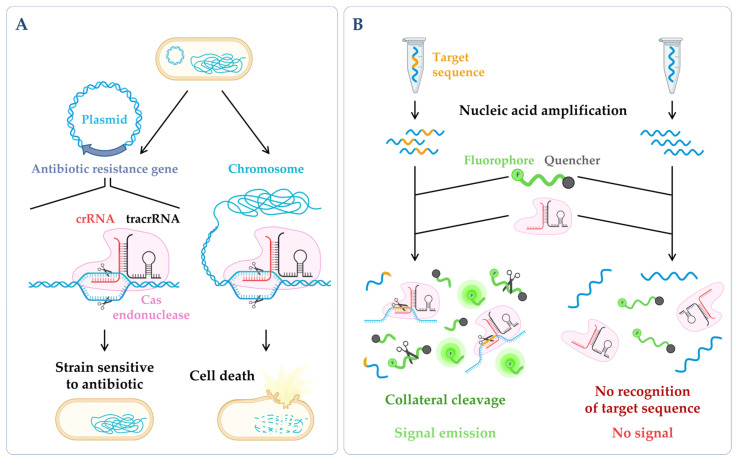
**CRISPR technologies.** (**A**) *CRISPR-based antimicrobials.* The system has been successfully tested through the directed degradation of the antibiotic resistance gene located in a plasmid (left side) leading to the recovery of the bacterial antibiotic sensitivity or the directed degradation of chromosomal genes, and consequently, cell death (bactericidal) [186]. (**B)**
*CRISPR-based diagnostics*. When CRISPR effector proteins (Cas) recognize the specific target site, their collateral cleavage capability is triggered (this indiscriminate nucleic acid cleavage only happens when the crRNA finds its match). The addition of a reporter, that only releases the signal upon cleavage, enables the emission of a signal that can be easily detected [187]. Figure created using BioRender.com (accessed on 11 November 2022).

## Data Availability

Not applicable.

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
