# Peer review of "Developing New Tools to Fight Human Pathogens: A Journey through the Advances in RNA Technologies"

_microorganisms, 2022, doi:10.3390/microorganisms10112303_

Round 1

Reviewer 1 Report

The review by Costa et al. provides an up-to-date overview of microbial RNA metabolism and describes advances in RNA-based tools and technologies that could be or are already being used to develop antimicrobial therapeutics. It is a neat idea to combine the two aspects of the RNA field, but a stronger connection would be desirable. Currently, metabolism and technological advances are treated separately. Considering that this is a relatively comprehensive review of the literature, the addition of figures to both parts would be appreciated (both to illustrate the text and to link the two parts).

Major comments:

Abstract: The sentence on lines 13-14 is unusual; I suggest rephrasing for clarity. In line 16, “we do an update” is perhaps too colloquial. Abbreviation “ncRNA” is used without its full name.

Introduction: Some sentences are either well known, or self-explanatory (e.g. line 30, line 33-34). The pathways in which the RNA possesses a regulatory role should be illustrated with a figure (lines 52-60).

Ribonuclease section: The part with viral RNAses would likely be a separate paragraph (line 107). I suggest the authors illustrate this part with a figure too. For instance, a specific example how some RNAses participate in the pathogenesis (both viral and bacterial).

Small ncRNA section: This section would benefit from a figure too. Because ncRNAs regulate the levels of various outer membrane proteins (OmpD in Salmonella, OmpF in E. coli, OmpC in Shigella), a single figure may illustrate different individual cases. Another potential figure for this part of the review would be one showing use of synthetic biology tools to target ncRNA pathways (a specific example). It is also not clear how the vaccine mentioned in lines 190-191 works.

RNA thermometers and riboswitches: the use of the RNA thermometers in reference 130 may be best illustrated with a figure (lines 343-352). Same goes for examples of use of riboswitches mentioned in lines 371-378.

CRISPR section: Some parts are well known and likely unnecessary (e.g. lines 483-489).

RNA technology: A figure or a table listing the benefits of different techniques would be desirable. The authors should also incorporate the advances in RNA sequencing using nanopores, and also the detection of posttranscriptional RNA modifications using that technology.

ASO section: Please illustrate the mechanism with a figure. Phosphorothioates are mistakenly capitalized (line 648).

Minor comments:

-some words are capitalized for no apparent reason, e.g. synthetic biology, RNA thermometers.

Author Response

We were glad to hear the positive feedback from the reviewer and we have tried to address his suggestions, namely regarding the inclusion of new figures in the manuscript. As such, we have now included three new figures in the manuscript that we think that meets the requirements of the reviewer. We also have corrected the words that were mistakenly capitalized and removed the redundant parts of the text.

Reviewer 2 Report

The authors review and discuss the emerging technologies on the RNA field. Overall, the review as presented is a significant work and has paving the way for the solid establishment of RNA-based therapies in the future. However, there are some concerns should be addressed and described as below.

1.      This manuscript is too long and redundancy. Please check the article throughout and simplify contents, such as Ribonucleases part.

2.      Suggest to supplement the research progress of siRNA as therapeutic drugs.

3.      In the figures1, please update the landmarks on RNA technologies to 2022.

4.      In RNA technology part, the authors need to furtherly discuss how RNA-techniques be optimized for diagnosis and therapeutics of diseases.

Author Response

We were very pleased to know that the referee found our work scientifically sound and useful. Regarding the concerns pointed out by the referee, we have tried to remove the redundancies of the text; we have included a new section in the manuscript regarding the progresses of siRNAs as therapeutic drugs; we also have updated the landmarks shown in figure 1.

Concerning point 4, in the RNA technologies section we first describe how the different variants of RNA sequencing have enabled the progresses in the identification of several microorganisms and characterization of their mechanisms of infection. Then, we describe the use of antisense oligonucleotides (ASOs) as alternative antimicrobial therapeutics and, finally we present how mRNA vaccines work and their current applications. Since we have now included in this revised version the description of the use of siRNAs as a therapeutic drug, and for the sake of manuscript size, we think that the content is appropriate.

Reviewer 3 Report

The authors have provided a detailed and comprehensive review of various RNA technologies that are used to fight the human pathogens. These include small non-coding RNA, RNA-seq, CRISPR system. Overall, the review is well written and the authors have given all the necessary citations/reference. I enjoyed reading it and should this review will be of interested to readers who are studying host-pathogen interactions.

Author Response

We acknowledge all the positive comments of the referee and we are very happy with it.

Round 2

Reviewer 1 Report

The revised manuscript is sufficiently improved.